# CONNECTING THE DOTS BETWEEN MLE AND RL FOR SEQUENCE GENERATION

**Bowen Tan\*, Zhiting Hu\*, Zichao Yang, Ruslan Salakhutdinov, Eric P. Xing**
(\*Equal contribution)
Carnegie Mellon University & Petuum Inc.
`{bwkevintan,zhitinghu,yangtze2301}@gmail.com rsalakhu@cs.cmu.edu`
`eric.xing@petuum.com`

## ABSTRACT

Sequence generation models such as recurrent networks can be trained with a diverse set of learning algorithms. For example, maximum likelihood learning is simple and efficient, yet suffers from the exposure bias problem. Reinforcement learning like policy gradient addresses the problem but can have prohibitively poor exploration efficiency. A variety of other algorithms such as RAML, SPG, and data noising, have also been developed from different perspectives. This paper establishes a formal connection between these algorithms. We present a generalized entropy regularized policy optimization formulation, and show that the apparently divergent algorithms can all be reformulated as special instances of the framework, with the only difference being the configurations of reward function and a couple of hyperparameters. The unified interpretation offers a systematic view of the varying properties of exploration and learning efficiency. Besides, based on the framework, we present a new algorithm that dynamically interpolates among the existing algorithms for improved learning. Experiments on machine translation and text summarization demonstrate the superiority of the proposed algorithm.

## 1 INTRODUCTION

Sequence generation is a ubiquitous problem in many applications, such as machine translation (Wu et al., 2016; Sutskever et al., 2014), text summarization (Hovy & Lin, 1998; Rush et al., 2015), image captioning (Vinyals et al., 2015; Karpathy & Fei-Fei, 2015), and so forth. Great advances in these tasks have been made by the development of sequence models such as recurrent neural networks (RNNs) with different cells (Hochreiter & Schmidhuber, 1997; Chung et al., 2014) and attention mechanisms (Bahdanau et al., 2015; Luong et al., 2015). These models can be trained with a variety of learning algorithms.

The standard training algorithm is based on maximum-likelihood estimation (MLE) which seeks to maximize the log-likelihood of ground-truth sequences. Despite the computational simplicity and efficiency, MLE training suffers from the *exposure bias* (Ranzato et al., 2016). That is, the model is trained to predict the next token given the previous ground-truth tokens; while at test time, since the resulting model does not have access to the ground truth, tokens generated by the model itself are instead used to make the next prediction. This discrepancy between training and test leads to the issue that mistakes in prediction can quickly accumulate. Recent efforts have been made to alleviate the issue, many of which resort to the reinforcement learning (RL) techniques (Ranzato et al., 2016; Bahdanau et al., 2017; Ding & Soricut, 2017). For example, Ranzato et al. (2016) adopt policy gradient (Sutton et al., 2000) that avoids the training/test discrepancy by using the same decoding strategy. However, RL-based approaches for sequence generation can face challenges of prohibitively poor sample efficiency and high variance. For more practical training, a diverse set of methods has been developed that are in a middle ground between the two paradigms of MLE and RL. For example, RAML (Norouzi et al., 2016) adds reward-aware perturbation to the MLE data examples; SPG (Ding & Soricut, 2017) leverages reward distribution for effective sampling of policy gradient. Other approaches such as data noising (Xie et al., 2017) also show improved results.

In this paper, we establish a unified perspective of the broad set of learning algorithms. Specifically, we present a generalized entropy regularized policy optimization framework, and show that the apparently diverse algorithms, such as MLE, RAML, SPG, and data noising, can all be re-formulated as special instances of the framework, with the only difference being the choice of reward and the values of a couple of hyperparameters (Figure 1). In particular, we show MLE is equivalent to using a *delta*-function reward that assigns 1 to samples that exactly match data examples while $-\infty$ to any other samples. Such extremely restricted reward has literally disabled any exploration of the model beyond training data, yielding the exposure bias. Other algorithms essentially use rewards that are more smooth, and also leverage model distribution for exploration, which generally results in a larger effective exploration space, more difficult training, and better test-time performance.

Besides the new understandings of the existing algorithms, the unified perspective also facilitates to develop new algorithms for improved learning. We present an example new algorithm that, as training proceeds, gradually expands the exploration space by annealing the reward and hyperparameter values. The annealing in effect dynamically interpolates among the existing algorithms. Experiments on machine translation and text summarization show the interpolation algorithm achieves significant improvement over the various existing methods.

## 2 RELATED WORK

Sequence generation models are usually trained to maximize the log-likelihood of data by feeding the ground-truth tokens during decoding. Reinforcement learning (RL) addresses the discrepancy between training and test by also using models' own predictions at training time. Various RL approaches have been applied for sequence generation, such as policy gradient (Ranzato et al., 2016) and actor-critic (Bahdanau et al., 2017). Softmax policy gradient (SPG) (Ding & Soricut, 2017) additionally incorporates the reward distribution to generate high-quality sequence samples. The algorithm is derived by applying a log-softmax trick to adapt the standard policy gradient objective. Reward augmented maximum likelihood (RAML) (Norouzi et al., 2016) is an algorithm in between MLE and policy gradient. It is originally developed to go beyond the maximum likelihood criteria and incorporate task metric (such as BLEU for machine translation) to guide the model learning. Mathematically, RAML shows that MLE and maximum-entropy policy gradient are respectively minimizing KL divergences in opposite directions. We reformulate both SPG and RAML in a new perspective, and show they are precisely instances of a general entropy regularized policy optimization framework. The new framework provides a more principled formulation for both algorithms. Besides the algorithms discussed in the paper, there are other learning methods for sequence models. For example, Hal Daumé et al. (2009); Leblond et al. (2018); Wiseman & Rush (2016) use a learning-to-search paradigm for sequence generation or structured prediction. Scheduled Sampling (Bengio et al., 2015) adapts MLE by randomly replacing ground-truth tokens with model predictions as the input for decoding the next-step token. Our empirical comparison shows improved performance of the proposed algorithm.

Policy optimization for reinforcement learning is studied extensively in robotics and game environment. For example, Peters et al. (2010) introduce a relative entropy regularization to reduce information loss during learning. Schulman et al. (2015) develop a trust-region approach for monotonic improvement. Dayan & Hinton (1997); Levine (2018); Abdolmaleki et al. (2018) study the policy optimization algorithms in a probabilistic inference perspective. The entropy-regularized policy optimization formulation presented here can be seen as a generalization of many of the previous policy optimization methods, as shown in the next section. Besides, we formulate the framework in the sequence generation context.

## 3 CONNECTING THE DOTS

We first present a *generalized* formulation of an entropy regularized policy optimization framework, to which a broad set of learning algorithms for sequence generation are connected. In particular, we show the conventional maximum likelihood learning is a special case of the policy optimization formulation. This provides new understandings of the exposure bias problem as well as the exploration efficiency of the algorithms. We further show that the framework subsumes as special cases other well-known

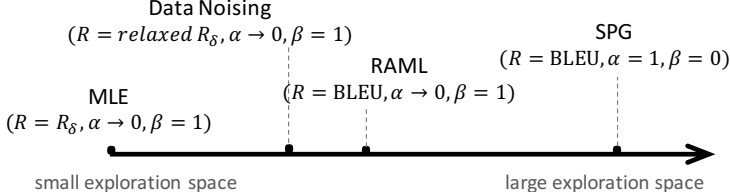

Figure 1: A unified formulation of different learning algorithms. Each algorithm is a special instance of the general ERPO framework taking certain specifications of the hyperparameters $(R, \alpha, \beta)$ (Eq.1).

learning methods that were originally developed in diverse perspectives. We thus establish a unified, principled view of the broad class of works.

Let us first set up the notations for the sequence generation setting. Let $\boldsymbol{x}$ be the input and $\boldsymbol{y} = (y_1, \ldots, y_T)$ the sequence of $T$ tokens in the target space. For example, in machine translation, $\boldsymbol{x}$ is the sentence in source language and $\boldsymbol{y}$ is in target language. Let $(\boldsymbol{x}, \boldsymbol{y}^*)$ be a training example drawn from the empirical data distribution, where $\boldsymbol{y}^*$ is the ground truth sequence. We aim to learn a sequence generation model $p_\theta(\boldsymbol{y}|\boldsymbol{x}) = \prod_t p_\theta(y_t|\boldsymbol{y}_{1:t-1}, \boldsymbol{x})$ parameterized with $\boldsymbol{\theta}$. The model can, for example, be a recurrent network. It is worth noting that though we present in the sequence generation context, the formulations can straightforwardly be extended to other settings such as robotics and game environment.

## 3.1 ENTROPY REGULARIZED POLICY OPTIMIZATION (ERPO)

Policy optimization is a family of reinforcement learning (RL) algorithms that seeks to learn the parameter $\boldsymbol{\theta}$ of the model $p_\theta$ (a.k.a policy). Given a reward function $R(\boldsymbol{y}|\boldsymbol{y}^*) \in \mathbb{R}$ (e.g., BLEU score in machine translation) that evaluates the quality of generation $\boldsymbol{y}$ against the true $\boldsymbol{y}^*$, the general goal of policy optimization is to maximize the expected reward. A rich research line of *entropy regularized* policy optimization (ERPO) stabilizes the learning by augmenting the objective with information theoretic regularizers. Here we present a generalized formulation of ERPO. Assuming a general distribution $q(\boldsymbol{y}|\boldsymbol{x})$ (more details below), the objective we adopt is written as

$$\mathcal{L}(q, \boldsymbol{\theta}) = \mathbb{E}_q\left[R(\boldsymbol{y}|\boldsymbol{y}^*)\right] - \alpha\text{KL}\big(q(\boldsymbol{y}|\boldsymbol{x})\|p_\theta(\boldsymbol{y}|\boldsymbol{x})\big) + \beta\text{H}(q), \tag{1}$$

where $\text{KL}(\cdot\|\cdot)$ is the Kullback–Leibler divergence forcing $q$ to stay close to $p_\theta$; $\text{H}(\cdot)$ is the Shannon entropy imposing maximum entropy assumption on $q$; and $\alpha$ and $\beta$ are balancing weights of the respective terms. In the RL literature, the distribution $q$ has taken various forms, leading to different policy optimization algorithms. For example, setting $q$ to a non-parametric policy and $\beta = 0$ results in the prominent relative entropy policy search (Peters et al., 2010) algorithm. Assuming $q$ as a parametric distribution and $\alpha = 0$ leads to the commonly-used maximum entropy policy gradient (Ziebart, 2010; Haarnoja et al., 2017). Letting $q$ be a variational distribution and $\beta = 0$ corresponds to the probabilistic inference formulation of policy gradient (Abdolmaleki et al., 2018; Levine, 2018). Related objectives have also been used in other popular RL algorithms (Schulman et al., 2015; 2017; Teh et al., 2017).

We assume a non-parametric $q$. The above objective can be maximized with an EM-style procedure that iterates two coordinate ascent steps optimizing $q$ and $\boldsymbol{\theta}$, respectively. At iteration $n$:

$$\begin{aligned}\text{E-step:} \quad & q^{n+1}(\boldsymbol{y}|\boldsymbol{x}) \propto \exp\left\{\frac{\alpha \log p_{\theta^n}(\boldsymbol{y}|\boldsymbol{x}) + R(\boldsymbol{y}|\boldsymbol{y}^*)}{\alpha + \beta}\right\}, \\ \text{M-step:} \quad & \boldsymbol{\theta}^{n+1} = \arg\max_\theta \mathbb{E}_{q^{n+1}}\big[\log p_\theta(\boldsymbol{y}|\boldsymbol{x})\big].\end{aligned} \tag{2}$$

The E-step is obtained with simple Lagrange multipliers. Note that $q$ has a closed-form solution in the E-step. We can have an intuitive interpretation of its form. First, it is clear to see that if $\alpha \to \infty$, we have $q^{n+1} = p_\theta^n$. This is also reflected in the objective Eq.(1) where the weight $\alpha$ encourages $q$ to be close to $p_\theta$. Second, the weight $\beta$ serves as the temperature of the $q$ softmax distribution. In particular, a large temperature $\beta \to \infty$ makes $q$ a uniform distribution, which is consistent to the outcome of an infinitely large maximum entropy regularization in Eq.(1). In terms of the M-step, the update rule can be interpreted as maximizing the log-likelihood of samples from the distribution $q$.

In the context of sequence generation, it is sometimes more convenient to express the equations at token level, as shown shortly. To this end, we decompose $R(\boldsymbol{y}|\boldsymbol{y}^*)$ along the time steps:

$$R(\boldsymbol{y}|\boldsymbol{y}^*) = \sum_t R(\boldsymbol{y}_{1:t}|\boldsymbol{y}^*) - R(\boldsymbol{y}_{1:t-1}|\boldsymbol{y}^*) := \sum_t \Delta R(y_t|\boldsymbol{y}_{1:t-1}, \boldsymbol{y}^*), \quad (3)$$

where $\Delta R(y_t|\boldsymbol{y}^*, \boldsymbol{y}_{1:t-1})$ measures the reward contributed by token $y_t$. The solution of $q$ in Eq.(2) can then be re-written as:

$$q^{n+1}(\boldsymbol{y}|\boldsymbol{x}) \propto \prod_t \exp\left\{ \frac{\alpha \log p_{\theta^n}(y_t|\boldsymbol{y}_{1:t-1}, \boldsymbol{x}) + \Delta R(y_t|\boldsymbol{y}_{1:t-1}, \boldsymbol{y}^*)}{\alpha + \beta} \right\} \quad (4)$$

The above ERPO framework has three key hyperparameters, namely $(R, \alpha, \beta)$. In the following, we show that different values of the three hyperparameters correspond to different learning algorithms (Figure 1). We first connect MLE to the above general formulation, and compare and discuss the properties of MLE and regular ERPO from the new perspective.

## 3.2 MLE as a Special Case of ERPO

Maximum likelihood estimation is the most widely-used approach to learn a sequence generation model due to its simplicity and efficiency. It aims to find the optimal parameter value that maximizes the data log-likelihood:

$$\boldsymbol{\theta}^* = \arg\max_\theta \mathcal{L}_{\text{MLE}}(\boldsymbol{\theta}) = \arg\max_\theta \log p_\theta(\boldsymbol{y}^*|\boldsymbol{x}). \quad (5)$$

As discussed in section 1, MLE suffers from the exposure bias problem as the model is only exposed to the training data, rather than its own predictions, by using the ground-truth subsequence $\boldsymbol{y}_{1:t-1}^*$ to evaluate the probability of $y_t^*$.

We show that the MLE objective can be recovered from Eq.(2) with specific reward and weight configurations. Consider a $\delta$-reward defined as[1]:

$$R_\delta(\boldsymbol{y}|\boldsymbol{y}^*) = \begin{cases} 1 & \text{if } \boldsymbol{y} = \boldsymbol{y}^* \\ -\infty & \text{otherwise.} \end{cases} \quad (6)$$

Let $(R = R_\delta, \alpha \to 0, \beta = 1)$. From the E-step of Eq.(2), we have $q(\boldsymbol{y}|\boldsymbol{x}) = 1$ if $\boldsymbol{y} = \boldsymbol{y}^*$ and $0$ otherwise. The M-step is therefore equivalent to $\arg\max_\theta \log p_\theta(\boldsymbol{y}^*|\boldsymbol{x})$, which recovers precisely the MLE objective in Eq.(5).

That is, MLE can be seen as an instance of the policy optimization algorithm with the $\delta$-reward and the above weight values. Any sample $\boldsymbol{y}$ that fails to match precisely the data $\boldsymbol{y}^*$ will receive a negative infinite reward and never contribute to model learning.

**Exploration efficiency**

The ERPO reformulation of MLE provides a new statistical explanation of the exposure bias problem. Specifically, a very small $\alpha$ value makes the model distribution ignored during sampling from $q$, while the $\delta$-reward permits only samples that match training examples. The two factors in effect make void any exploration beyond the small set of training data (Figure 2(a)), leading to a brittle model that performs poorly at test time due to the extremely restricted exploration. On the other hand, however, a key advantage of the $\delta$-reward specification is that its regular reward shape allows extreme pruning of the huge sample space, resulting in a space that includes exactly the training examples. This makes the MLE implementation very simple and the computation very efficient in practice.

On the contrary, common rewards (e.g., BLEU) used in policy optimization are more smooth than the $\delta$-reward, and permit exploration in a broader space. However, such rewards usually do not have a regular shape as the $\delta$-reward, and thus are not amenable to sample space pruning. Generally, a larger exploration space would lead to a harder training problem. Also, when it comes to the huge sample space, the rewards are still very sparse (e.g., most sequences have BLEU=0 against a reference sequence). Such reward sparsity can make exploration inefficient and even impractical.

---

[1]For token-level, define $R_\delta(\boldsymbol{y}_{1:t}|\boldsymbol{y}^*) = t/T^*$ if $\boldsymbol{y}_{1:t} = \boldsymbol{y}_{1:t}^*$ and $-\infty$ otherwise, where $T^*$ is the length of $\boldsymbol{y}^*$. Note that the $R_\delta$ value of $\boldsymbol{y} = \boldsymbol{y}^*$ can also be set to any constant larger than $-\infty$.

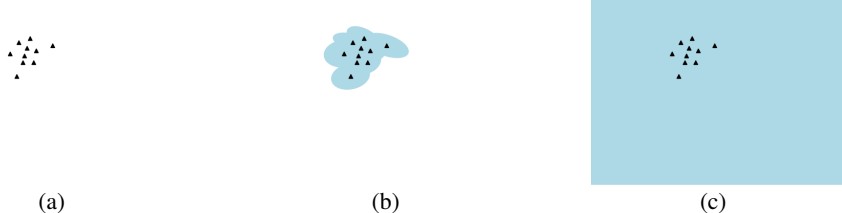

(a)         (b)         (c)

Figure 2: Effective exploration space of different algorithms. **(a)**: The exploration space of MLE is exactly the set of training examples. **(b)**: RAML and Data Noising use smooth rewards and allow larger exploration space surrounding the training examples. **(c)**: Common policy optimization such as SPG basically allows the whole exploration space.

Given the opposite algorithm behaviors in terms of exploration and computation efficiency, it is a natural idea to seek a middle ground between the two extremes to combine the advantages of both. A broad set of such approaches have been recently developed. We re-visit some of the popular ones, and show that these apparently divergent approaches can all be reformulated within our ERPO framework (Eqs.1-4) with varying reward and weight specifications.

### 3.3 Reward-Augmented Maximum Likelihood (RAML)

RAML (Norouzi et al., 2016) was originally proposed to incorporate task metric reward into the MLE training, and has shown superior performance to the vanilla MLE. Specifically, it introduces an exponentiated reward distribution $e(\boldsymbol{y}|\boldsymbol{y}^*) \propto \exp\{R(\boldsymbol{y}|\boldsymbol{y}^*)\}$ where $R$, as in vanilla policy optimization, is a task metric such as BLEU. RAML maximizes the following objective:

$$\mathcal{L}_{\text{RAML}}(\boldsymbol{\theta}) = \mathbb{E}_{\boldsymbol{y} \sim e(\boldsymbol{y}|\boldsymbol{y}^*)}\big[\log p_{\theta}(\boldsymbol{y}|\boldsymbol{x})\big]. \tag{7}$$

That is, unlike MLE that directly maximizes the data log-likelihood, RAML first perturbs the data proportionally to the reward distribution $e$, and maximizes the log-likelihood of the resulting samples.

The RAML objective reduces to the vanilla MLE objective if we replace the task reward $R$ in $e(\boldsymbol{y}|\boldsymbol{y}^*)$ with the MLE $\delta$-reward (Eq.6). The relation between MLE and RAML still holds within our new formulation (Eqs.1-2). In particular, similar to how we recovered MLE from Eq.(2), let $(\alpha \to 0, \beta = 1)^2$, but set $R$ to the task metric reward, then the M-step of Eq.(2) is precisely equivalent to maximizing the above RAML objective.

Formulating within the same framework allows us to have an immediate comparison between RAML and others. In particular, compared to MLE, the use of smooth task metric reward $R$ instead of $R_{\delta}$ permits a larger effective exploration space surrounding the training data (Figure 2(b)), which helps to alleviate the exposure bias problem. On the other hand, $\alpha \to 0$ as in MLE still limits the exploration as it ignores the model distribution. Thus, RAML takes a step from MLE toward regular RL, and has effective exploration space size and exploration efficiency in between.

### 3.4 Softmax Policy Gradient (SPG)

SPG (Ding & Soricut, 2017) was developed in the perspective of adapting the vanilla policy gradient (Sutton et al., 2000) to use reward for sampling. SPG has the following objective:

$$\mathcal{L}_{SPG}(\boldsymbol{\theta}) = \log \mathbb{E}_{p_{\theta}}\left[\exp R(\boldsymbol{y}|\boldsymbol{y}^*)\right], \tag{8}$$

where $R$ is a common reward as above. As a variant of the standard policy gradient algorithm, SPG aims to address the exposure bias problem and shows promising results (Ding & Soricut, 2017).

We show SPG can readily fit into our ERPO framework. Specifically, taking gradient of Eq.(8) w.r.t $\boldsymbol{\theta}$, we immediately get the same update rule as in Eq.(2) with ($\alpha = 1, \beta = 0, R =$ common reward).

Note that the only difference between the SPG and RAML configuration is that now $\alpha = 1$. SPG thus moves a step further than RAML by leveraging both the reward and the model distribution for full

---

[2]The exponentiated reward distribution $e$ can also include a temperature $\tau$ (Norouzi et al., 2016). In this case, we set $\beta = \tau$.

exploration (Figure 2(c)). Sufficient exploration at training time would in theory boost the test-time performance. However, with the increased learning difficulty, additional sophisticated optimization and approximation techniques have to be used (Ding & Soricut, 2017) to make the training practical.

## 3.5 DATA NOISING

Adding noise to training data is a widely adopted technique for regularizing models. Previous work (Xie et al., 2017) has proposed several data noising strategies in the sequence generation context. For example, a *unigram noising*, with probability $\gamma$, replaces each token in data $\boldsymbol{y}^*$ with a sample from the unigram frequency distribution. The resulting noisy data is then used in MLE training.

Though previous literature has commonly seen such techniques as a data pre-processing step that differs from the above learning algorithms, we show the ERPO framework can also subsume data noising as a special instance. Specifically, starting from the ERPO reformulation of MLE which takes $(R = R_\delta, \alpha \to 0, \beta = 1)$ (section 3.2), data noising can be formulated as using a locally relaxed variant of $R_\delta$. For example, assume $\boldsymbol{y}$ has the same length with $\boldsymbol{y}^*$ and let $\Delta_{\boldsymbol{y}, \boldsymbol{y}^*}$ be the set of tokens in $\boldsymbol{y}$ that differ from the corresponding tokens in $\boldsymbol{y}^*$, then a simple data noising strategy that randomly replaces a single token $y_t^*$ with another uniformly picked token is equivalent to using a reward $R_\delta'(\boldsymbol{y}|\boldsymbol{y}^*)$ that takes 1 when $|\Delta_{\boldsymbol{y}, \boldsymbol{y}^*}| = 1$ and $-\infty$ otherwise. Likewise, the above unigram noising (Xie et al., 2017) is equivalent to using a reward

$$R_\delta^{\text{unigram}}(\boldsymbol{y}|\boldsymbol{y}^*) = \begin{cases} \log\left(\gamma^{|\Delta_{\boldsymbol{y}, \boldsymbol{y}^*}|}(1-\gamma)^{T-|\Delta_{\boldsymbol{y}, \boldsymbol{y}^*}|} \prod_{y_t \in \Delta_{\boldsymbol{y}, \boldsymbol{y}^*}} u(y_t)\right) & \text{if } T = T^* \\ -\infty & \text{otherwise,} \end{cases} \tag{9}$$

where $u(\cdot)$ is the unigram frequency distribution.

With a relaxed (i.e., smoothed) reward, data noising expands the exploration space of vanilla MLE locally (Figure 2(b)). The effect is essentially the same as the RAML algorithm (section 3.3), except that RAML expands the exploration space based on the task metric reward.

**Other Algorithms** Ranzato et al. (2016) made an early attempt to address the exposure bias problem by exploiting the classic policy gradient algorithm (Sutton et al., 2000) and mixing it with MLE training. We show in the supplementary materials that the algorithm is closely related to the ERPO framework, and can be recovered with moderate approximations. Section 2 discusses more relevant algorithms for sequence generation learning.

## 4  INTERPOLATION ALGORITHM

We have presented the generalized ERPO framework, and connected a series of well-used learning algorithms by showing that they are all instances of the framework with certain specifications of the three hyperparameters $(R, \alpha, \beta)$. Each of the algorithms can be seen as a point in the hyperparameter space (Figure 1). Generally, a point with a more restricted reward function $R$ and a very small $\alpha$ tends to have a smaller effective exploration space and allow efficient learning (e.g., MLE), while in contrast, a point with smooth $R$ and a larger $\alpha$ would lead to a more difficult learning problem, but permit more sufficient exploration and better test-time performance (e.g., (softmax) policy gradient). The unified perspective provides new understandings of the existing algorithms, and also facilitates to develop new algorithms for further improvement. Here we present an example algorithm that interpolates the existing ones.

The *interpolation* algorithm exploits the natural idea of starting learning from the most restricted yet easiest problem configuration, and gradually expands the exploration space to reduce the discrepancy from the test time. The easy-to-hard learning paradigm resembles the curriculum learning (Bengio et al., 2009). As we have mapped the algorithms to points in the hyperparameter space, interpolation becomes very straightforward, which requires only *annealing* of the hyperparameter values.

Specifically, in the general update rules Eq.(2), we would like to anneal from using $R_\delta$ to using smooth common reward, and anneal from exploring by only $R$ to exploring by both $R$ and $p_\theta$. Let $R_{\text{comm}}$ denote a common reward (e.g., BLEU). The interpolated reward can be written in the form $R = \lambda R_{\text{comm}} + (1 - \lambda)R_\delta$, for $\lambda \in [0, 1]$. Plugging $R$ into $q$ in Eq.(2) and re-organizing the scalar weights, we obtain the numerator of $q$ in the form: $c \cdot (\lambda_1 \log p_\theta + \lambda_2 R_{\text{comm}} + \lambda_3 R_\delta)$, where

| Model | BLEU |
|---|---|
| MLE | $26.44 \pm 0.18$ |
| RAML (Norouzi et al., 2016) | $27.22 \pm 0.14$ |
| SPG (Ding & Soricut, 2017) | $26.62 \pm 0.05$ |
| MIXER (Ranzato et al., 2016) | $26.53 \pm 0.11$ |
| Scheduled Sampling (Bengio et al., 2015) | $26.76 \pm 0.17$ |
| Ours | $\mathbf{27.82 \pm 0.11}$ |

Table 1: Results of machine translation.

$(\lambda_1, \lambda_2, \lambda_3)$ is defined as a distribution (i.e., $\lambda_1 + \lambda_2 + \lambda_3 = 1$), and, along with $c \in \mathbb{R}$, are determined by $(\alpha, \beta, \lambda)$. For example, $\lambda_1 = \alpha/(\alpha + 1)$. We gradually increase $\lambda_1$ and $\lambda_2$ and decrease $\lambda_3$ as the training proceeds.

Further, noting that $R_\delta$ is a Delta function (Eq.6) which would make the above direct function interpolation problematic, we borrow the idea from the Bayesian *spike-and-slab* factor selection method (Ishwaran et al., 2005). That is, we introduce a categorical random variable $z \in \{1, 2, 3\}$ that follows the distribution $(\lambda_1, \lambda_2, \lambda_3)$, and augment $q$ as $q(\boldsymbol{y}|\boldsymbol{x}, z) \propto \exp\{c \cdot (\mathbb{1}(z = 1) \log p_\theta + \mathbb{1}(z = 2)R_{\text{comm}} + \mathbb{1}(z = 3)R_\delta)\}$. The M-step is then to maximize the objective with $z$ marginalized out: $\max_{\boldsymbol{\theta}} \mathbb{E}_{p(z)}\mathbb{E}_{q(\boldsymbol{y}|\boldsymbol{x},z)} [\log p_\theta(\boldsymbol{y}|\boldsymbol{x})]$. The spike-and-slab adaption essentially transforms the product of experts in $q$ to a mixture, which resembles the bang-bang rewarded SPG method (Ding & Soricut, 2017) where the name *bang-bang* refers to a system that switches abruptly between extreme states (i.e., the $z$ values). Finally, similar to (Ding & Soricut, 2017), we adopt the token-level formulation (Eq.4) and associate each token with a separate variable $z$.

We provide the pseudo-code of the interpolation algorithm in the supplements. It is notable that Ranzato et al. (2016) also develop an annealing strategy that mixes MLE and policy gradient training. As discussed in section 3 and the supplements, the algorithm can be seen as a special instance of the ERPO framework (with moderate approximation) we have presented. Next section shows improved performance of the proposed, more general algorithm compared to (Ranzato et al., 2016).

## 5 EXPERIMENTS

We evaluate the above interpolation algorithm in the tasks of machine translation and text summarization. The proposed algorithm consistently improves over a variety of previous methods.

Code will be released upon acceptance.

**Setup** In both tasks, we follow previous work (Norouzi et al., 2016; Ranzato et al., 2016) and use an attentional sequence-to-sequence model (Luong et al., 2015) where both the encoder and decoder are single-layer LSTM recurrent networks. The dimensions of word embedding, RNN hidden state, and attention are all set to 256. We apply dropout of rate 0.2 on the recurrent hidden state. We use Adam optimization for training, with an initial learning rate of 0.001 and batch size of 64. At test time, we use beam search decoding with a beam width of 5. Please see the supplementary materials for more configuration details.

### 5.1 MACHINE TRANSLATION

**Dataset** Our dataset is based on the common IWSLT 2014 (Cettolo et al., 2014) German-English machine translation data, as also used in previous evaluation (Norouzi et al., 2016; Ranzato et al., 2016). After proper pre-processing as described in the supplementary materials, we obtain the final dataset with train/dev/test size of around 146K/7K/7K, respectively. The vocabulary sizes of German and English are around 32K and 23K, respectively.

**Results** The BLEU metric (Papineni et al., 2002) is used as the reward and for evaluation. Table 1 shows the test-set BLEU scores of various methods. Besides the approaches described above, we also compare with the Scheduled Sampling method (Bengio et al., 2015) which combats the exposure bias by feeding model predictions at randomly-picked decoding steps during training. From the table, we can see the various approaches such as RAML provide improved performance over the vanilla

MLE, as more sufficient exploration is made at training time. Our proposed new algorithm performs best, as it interpolates among the existing algorithms to gradually increase the exploration space and solve the generation problem better.

Figure 3 shows the test-set BLEU scores against the training steps. We can see that, with annealing, our algorithm improves BLEU smoothly, and surpasses other algorithms to converge at a better point.

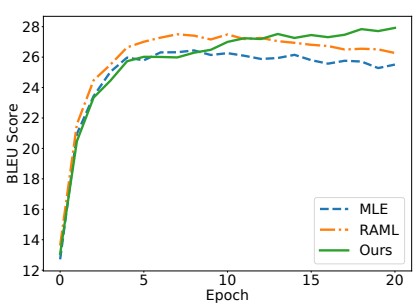

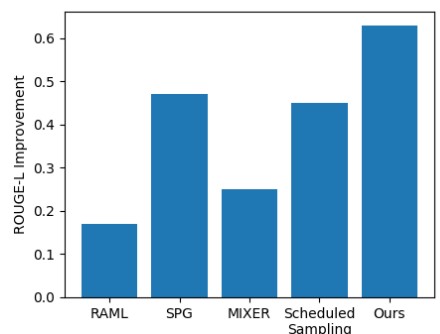

Figure 3: Convergence curve of learning algorithms in the task of machine translation. RAML is the second-best performing method (Table.1), inferior only to our algorithm.

Figure 4: Improvement on the ROUGE-L metric in comparison to MLE (e.g., RAML improves ROUGE-L by 0.17).

| Method | ROUGE-1 | ROUGE-2 | ROUGE-L |
|---|---|---|---|
| MLE | $36.11 \pm 0.21$ | $16.39 \pm 0.16$ | $32.32 \pm 0.19$ |
| RAML (Norouzi et al., 2016) | $36.30 \pm 0.04$ | $16.69 \pm 0.20$ | $32.49 \pm 0.17$ |
| SPG (Ding & Soricut, 2017) | $36.48 \pm 0.24$ | $16.84 \pm 0.26$ | $32.79 \pm 0.26$ |
| MIXER (Ranzato et al., 2016) | $36.34 \pm 0.23$ | $16.61 \pm 0.25$ | $32.57 \pm 0.15$ |
| Scheduled Sampling (Bengio et al., 2015) | $36.59 \pm 0.12$ | $16.79 \pm 0.22$ | $32.77 \pm 0.17$ |
| Ours | $\mathbf{36.72 \pm 0.29}$ | $\mathbf{16.99 \pm 0.17}$ | $\mathbf{32.95 \pm 0.33}$ |

Table 2: Results of text summarization.

## 5.2 TEXT SUMMARIZATION

**Dataset** We use the popular English Gigaword corpus (Graff et al., 2003) for text summarization, and pre-processed the data following (Rush et al., 2015). The resulting dataset consists of 200K/8K/2K source-target pairs in train/dev/test sets, respectively. More details are included in the supplements.

**Results** The ROUGE metrics (including -1, -2, and -L) (Lin, 2004) are the most commonly used metrics for text summarization. Following previous work (Ding & Soricut, 2017), we use the summation of the three ROUGE metrics as the reward in the learning algorithms. Table 2 show the results on the test set. The proposed interpolation algorithm achieves the best performance on all the three metrics. For easier comparison, Figure 4 shows the improvement of each algorithm compared to MLE in terms of ROUGE-L. The RAML algorithm, which performed well in machine translation, falls behind other algorithms in text summarization. In contrast, our method consistently provides the best results.

## 6 CONCLUSIONS

We have presented a unified perspective of a variety of well-used learning algorithms for sequence generation. The framework is based on a generalized entropy regularized policy optimization formulation, and we show these algorithms are mathematically equivalent to specifying certain hyperparameter configurations in the framework. The new principled treatment provides systematic understanding and comparison among the algorithms, and inspires further enhancement. The proposed interpolation algorithm shows consistent improvement in machine translation and text summarization. We would be excited to extend the framework to other settings such as robotics and game environments.

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

# A  POLICY GRADIENT & MIXER

Ranzato et al. (2016) made an early attempt to address the exposure bias problem by exploiting the policy gradient algorithm (Sutton et al., 2000). Policy gradient aims to maximizes the expected reward:

$$\mathcal{L}_{PG}(\boldsymbol{\theta}) = \mathbb{E}_{p_\theta}\left[R_{PG}(\boldsymbol{y}|\boldsymbol{y}^*)\right], \tag{10}$$

where $R_{PG}$ is usually a common reward function (e.g., BLEU). Taking gradient w.r.t $\boldsymbol{\theta}$ gives:

$$\nabla_\theta \mathcal{L}_{PG}(\boldsymbol{\theta}) = \mathbb{E}_{p_\theta}\left[R_{PG}(\boldsymbol{y}|\boldsymbol{y}^*)\nabla_\theta \log p_\theta(\boldsymbol{y}|\boldsymbol{x})\right]. \tag{11}$$

We now reveal the relation between the ERPO framework we present and the policy gradient algorithm. Starting from the M-step of Eq.(2) and setting $(\alpha = 1, \beta = 0)$ as in SPG (section 3.4), we use $p_{\theta^n}$ as the proposal distribution and obtain the importance sampling estimate of the gradient (we omit the superscript $n$ for notation simplicity):

$$\begin{aligned}
\mathbb{E}_q\left[\nabla_\theta \log p_\theta(\boldsymbol{y}|\boldsymbol{x})\right] &= \mathbb{E}_{p_\theta}\left[\frac{q(\boldsymbol{y}|\boldsymbol{x})}{p_\theta(\boldsymbol{y}|\boldsymbol{x})}\nabla_\theta \log p_\theta(\boldsymbol{y}|\boldsymbol{x})\right] \\
&= 1/Z_\theta \cdot \mathbb{E}_{p_\theta}\left[\exp\{R(\boldsymbol{y}|\boldsymbol{y}^*)\} \cdot \nabla_\theta \log p_\theta(\boldsymbol{y}|\boldsymbol{x})\right],
\end{aligned} \tag{12}$$

where $Z_\theta = \int_y \exp\{\log p_\theta + R\}$ is the normalization constant of $q$, which can be considered as adjusting the step size of gradient descent.

We can see that Eq.(12) recovers Eq.(11) if we further set $R = \log R_{PG}$, and omit the scaling factor $Z_\theta$. In other words, policy gradient can be seen as a special instance of the general ERPO framework with $(R = \log R_{PG}, \alpha = 1, \beta = 0)$ and with $Z_\theta$ omitted.

The **MIXER** algorithm (Ranzato et al., 2016) incorporates an annealing strategy that mixes between MLE and policy gradient training. Specifically, given a ground-truth example $\boldsymbol{y}^*$, the first $m$ tokens $\boldsymbol{y}^*_{1:m}$ are used for evaluating MLE loss, and starting from step $m + 1$, policy gradient objective is used. The $m$ value decreases as training proceeds. With the relation between policy gradient and ERPO as established above, MIXER can be seen as a specific instance of the proposed interpolation algorithm (section 4) that follows a restricted annealing strategy for token-level hyperparameters $(\lambda_1, \lambda_2, \lambda_3)$. That is, for $t < m$ in Eq.4 (i.e.,the first $m$ steps), $(\lambda_1, \lambda_2, \lambda_3)$ is set to $(0, 0, 1)$ and $c = 1$, namely the MLE training; while for $t > m$, $(\lambda_1, \lambda_2, \lambda_3)$ is set to $(0.5, 0.5, 0)$ and $c = 2$.

# B  INTERPOLATION ALGORITHM

Algorithm 1 summarizes the interpolation algorithm described in section 4.

---
**Algorithm 1** Interpolation Algorithm

---
1: Initialize model parameter $\boldsymbol{\theta}$ and weights $\boldsymbol{\lambda} = (\lambda_1, \lambda_2, \lambda_3)$
2: **repeat**
3:     Get training example $(\boldsymbol{x}, \boldsymbol{y}^*)$
4:     **for** $t = 0, 1, \dots, T$ **do**
5:         Sample $z \in \{1, 2, 3\} \sim (\lambda_1, \lambda_2, \lambda_3)$
6:         **if** $z = 1$ **then**
7:             Sample token $y_t \sim \exp\{c \cdot \log p_\theta(y_t|\boldsymbol{y}_{1:t-1}, \boldsymbol{x})\}$
8:         **else if** $z = 2$ **then**
9:             Sample token $y_t \sim \exp\{c \cdot \Delta R_{comm}(y_t|\boldsymbol{y}_{1:t-1}, \boldsymbol{y}^*)\}$
10:         **else**
11:             Sample token $y_t \sim \exp\{c \cdot \Delta R_\delta\}$, i.e., set $y_t = y_t^*$
12:         **end if**
13:     **end for**
14:     Update $\boldsymbol{\theta}$ by maximizing the log-likelihood $\log p_\theta(\boldsymbol{y}|\boldsymbol{x})$
15:     Anneal $\boldsymbol{\lambda}$ by increasing $\lambda_1$ and $\lambda_2$ and decreasing $\lambda_3$
16: **until** convergence

---

## C EXPERIMENTAL SETTINGS

### C.1 DATA PRE-PROCESSING

For the machine translation dataset, we follow (Ma et al., 2017) for data pre-processing.

In text summarization, we sampled 200K out of the 3.8M pre-processed training examples provided by (Rush et al., 2015) for the sake of training efficiency. We used the refined validation and test sets provided by (Zhou et al., 2017).

### C.2 ALGORITHM SETUP

For RAML (Norouzi et al., 2016), we use the sampling approach (n-gram replacement) by (Ma et al., 2017) to sample from the exponentiated reward distribution. For each training example we draw 10 samples. The softmax temperature is set to $\tau = 0.4$.

For Scheduled Sampling (Bengio et al., 2015), the decay function we used is inverse-sigmoid decay. The probability of sampling from model $\epsilon_i = k/(k + \exp{(i/k)})$, where $k$ is a hyperparameter controlling the speed of convergence, which is set to $500$ and $600$ in the machine translation and text summarization tasks, respectively.

For MIXER (Ranzato et al., 2016), the advantage function we used for policy gradient is $R(\boldsymbol{y}_{1:T}|\boldsymbol{y}*) - R(\boldsymbol{y}_{1:m}|\boldsymbol{y}*)$.

For the proposed interpolation algorithm, we initialize the weights as $(\lambda_1, \lambda_2, \lambda_3) = (0.04, 0, 0.96)$, and increase $\lambda_1$ and $\lambda_2$ while decreasing $\lambda_3$ every time when the validation-set reward decreases. Specifically, we increase $\lambda_1$ by 0.06 once and increase $\lambda_2$ by 0.06 for four times, periodically. For example, at the first time the validation-set reward decreases, we increase $\lambda_1$, and at the second to fifth time, we increase $\lambda_2$, and so forth. The weight $\lambda_3$ is decreased by 0.06 every time we increase either $\lambda_1$ or $\lambda_2$. Notice that we would not update $\theta$ when the validation-set reward decreases.

## D ADDITIONAL RESULTS

Here we present additional results of machine translation using a dropout rate of 0.3 (Table 3). The improvement of the proposed interpolation algorithm over the baselines is comparable to that of using dropout 0.2 (Table 1 in the paper). For example, our algorithm improves over MLE by 1.5 BLEU points, and improves over the second best performing method RAML by 0.49 BLEU points. (With dropout 0.2 in Table 1, the improvements are 1.42 BLEU and 0.64, respectively.) We tested with dropout 0.5 and obtained similar results. The proposed interpolation algorithm outperforms existing approaches with a clear margin.

Figure 5 shows the convergence curves of the comparison algorithms.

| Model | BLEU |
|---:|:---|
| MLE | $26.63 \pm 0.11$ |
| RAML (Norouzi et al., 2016) | $27.64 \pm 0.09$ |
| SPG (Ding & Soricut, 2017) | $26.89 \pm 0.06$ |
| MIXER (Ranzato et al., 2016) | $27.00 \pm 0.13$ |
| Scheduled Sampling (Bengio et al., 2015) | $27.03 \pm 0.15$ |
| Ours | $\mathbf{28.13 \pm 0.12}$ |

Table 3: Results of machine translation when dropout is 0.3.

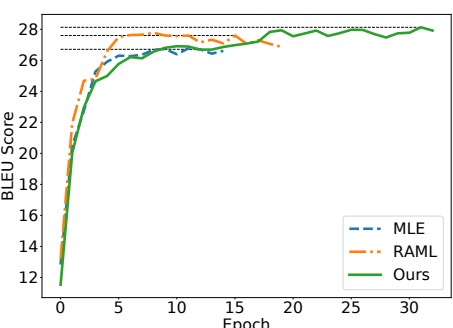

Figure 5: Convergence curve of learning algorithms in the task of machine translation with a dropout rate of 0.3. The horizontal dashed lines indicate the test-set results of each of the algorithms (reported in Table 3) picked according to the validation set performance.

