# OpenReview forum: "Connecting the Dots Between MLE and RL for Sequence Generation"
_ICLR.cc/2019/Workshop/drlStructPred — drlStructPred 2019_

### Official Review · AnonReviewer5 · 2019-04-05
**A generalized entropy regularized policy formulation encodes a variety of approaches to seq2seq learning, from maximum likelihood estimation to reinforcement learning implemented using the RAML, SPG, and data noising methods and leads to improved results on two tasks: machine translation and summarization.**

**Rating:** 5
**Confidence:** 2

**Review:**

This is a nice submission. The authors show how a generalized entropy regularized policy formulation encodes a variety of approaches to seq2seq learning, from maximum likelihood estimation to reinforcement learning implemented using the RAML, SPG, and data noising methods. Besides the theoretical insight, the authors show how by implementing an easy-to-hard paradigm that resembles curriculum learning leads to improvements on two tasks: machine translation and summarization.

The paper is likely to generate good discussions, especially with respect to other approaches to sequence learning that are not yet encompassed by the proposed framework.

---

### Official Review · AnonReviewer2 · 2019-04-05
**A useful unified formulation for several standard learning algorithms**

**Rating:** 3
**Confidence:** 2

**Review:**

The paper presents a unified objective function for optimizing sequence generation models. The unified formulation includes several standard learning algorithms as special cases. These standard algorithms include training by MLE, RAML, SPG, data noising, and sequential sampling.

The framework is based on policy optimization combined with entropy based regularization. The ground truth sequence y* is perturbed according to a probability distribution q(.|x) resulting in a perturbed target sequence y (i.e. y~q(.|x)). The goal is to maximize the expected reward under the new perturbed distribution of labels q. A KL-divergence penalty is also included to prevent deviation from the model distribution parameterized by theta. The objective function is regularized by imposing a maximum entropy assumption on q. The optimization objective is solved in an expectation maximization fashion. Different choices for the reward function and penalties for the KL-divergence and entropy terms yield standard learning algorithms ranging from MLE to SPG.

Finally, a new algorithm is proposed that dynamically interpolates between the different learning algorithms. The algorithm is evaluated in a very limited experimental setting for machine translation and text summarization.

Strength:
========

The unified formal connection is useful in understanding the difference between the learning algorithms in terms of how different target sequences are rewarded, which sequences are explored, and how the policy is regularized.

Weakness:
=========

The experimental setting for evaluating the new proposed algorithm is very limited. The dataset used for the machine translation experiment is relatively small. This is a major concern as the exposure bias problem can be mitigated by using larger datasets.


Clarity:
======

The paper is mostly clear. However, in section 3.1, it wasn’t clear at the beginning what the distribution q represents. The future reference in the “more details below” was not helpful. In general proving more intuition about the objection function in equation (1) would be helpful and make the presentation more clear.

---

### Official Review · AnonReviewer3 · 2019-04-06
**A nice review of recent approaches (RAML, SPG, data noising)**

**Rating:** 3
**Confidence:** 2

**Review:**

The authors unify several recent frameworks for training sequence generation models and claim that their general framework is principled and provides novel interpretations of previous algorithms. They propose an interpolated model which they evaluate on several sequence generation task and show improved performance.

It was unclear why the unified model was principled (this was not addressed in the paper), and I did not find that it provided novel interpretations beyond the existing literature. The unified objective amounts to adding previously used terms together with weights. The analysis of the objective is mostly qualitative and descriptive, rather than analytical. The paper could benefit from clearly defining terms like "regular shaped rewards", "smoothness" for discrete distributions, and "exploration area", etc.

The authors briefly mention learning to search approaches, but given their recent strong performance (e.g., Sabour et al. ICLR 2019), this is an important comparison that is missing.

For a workshop submission, the review of previous methods and empirical results are sufficiently interesting for acceptance.

---

### Official Review · AnonReviewer1 · 2019-04-06
**Nice contribution of unifying multiple common objectives in a single framework leading to algorithmic insight**

**Rating:** 5
**Confidence:** 2

**Review:**

This paper takes a number of widely-used algorithms for sequence generation, including maximum-likelihood, RAML, SPG and data noising and shows that they can all be viewed as optimizing a member of a family of objective functions, which is defined through three hyper-parameters that control parts of the objective - the reward, a maximum entropy term, and a KL term between a variational distribution and the model.

The authors show the exact values of the hyper-parameters that lead to these various objectives and also shed light onto how these hyper-parameters correspond to a trade-off between the exploration and difficulty of learning, where more exploration results in a more difficult learning problem.

Because now we have a family of objectives, the authors now naturally propose an algorithm that anneals the values of the hyper-parameters using a curriculum where simple learning without exploration happens at the beginning and more exploration is added later on to avoid local minima.

I found the analysis clear and interesting, the insights on the relation between the algorithms to be informative and the final simple algorithmic contribution to be natural and worthwhile. Overall, a nice paper that I think definitely fits well in this workshop.

It is worth noting that a similar but different attempt has been made in the context of sequence generation when there is no gold sequence given at training time: See Misra et al. 2018
http://www.cs.cornell.edu/~dkm/papers/mchy-emnlp.2018.pdf

---

### Official Review · AnonReviewer4 · 2019-04-07
**Good work, but not that natrual**

**Rating:** 3
**Confidence:** 3

**Review:**

The authors try to unify different algorithms for sequence generation and present a generalized entropy regularized policy optimization formulation. They show that divergent algorithms can be reformulated as special instances of the framework,
with the only difference being the configurations of reward function and a couple of hyperparameters.

The analysis and proposed framework are interesting.

I have several technical questions.

1. One cercen about the work is that the formulation in Eq. (1) is not natrual and it is not reasonable intuitively. Which one is the final model, q or \theta?  According to previous description, \theta is the model (or model parameters). However, for MLE, when \alpha->0, \theta will not make any impact to the reward defiend in Eq. (1), and only q determines the reward. That is, optimizing L(q,\theta) will only update q but not \theta.

2. "Generally, a larger exploration space would lead to a harder training problem." I don't get this point.  "common rewards (e.g., BLEU) used in policy optimization are more smooth than the -reward, and permit exploration in a broader space."  If BLEU is more smooth, why leads to a harder training problem?

---

### Decision · Program_Chairs · 2019-04-08
**Acceptance Decision**

**Decision:**

Accept

**Comment:**

This paper builds interesting connections of multiple methods (MLE, RL, RAML, etc) used to train model for sequence generation. All reviewers recommend acceptance.